# Xylanase Supplement Enhances the Growth Performance of Broiler by Modulating Serum Metabolism, Intestinal Health, Short-Chain Fatty Acid Composition, and Microbiota

**DOI:** 10.3390/ani14081182

**Published:** 2024-04-15

**Authors:** Xiaoli Wang, Danlei Li, Yibin Xu, Xiaoqing Ding, Shuang Liang, Lingyu Xie, Yongxia Wang, Xiuan Zhan

**Affiliations:** 1State Key Laboratory of Institute of Agro-Product Safety and Nutrition, Zhejiang Academy of Agricultural Sciences, Hangzhou 310021, China; wxlzaas@163.com; 2Key Laboratory of Animal Nutrition and Feed Science in East China, Ministry of Agriculture, College of Animal Sciences, Zhejiang University, Hangzhou 310058, China; lidanlitechi@163.com (D.L.); 21817002@zju.edu.cn (Y.X.); 21817065@zju.edu.cn (X.D.); 22117026@zju.edu.cn (S.L.); 22117016@zju.edu.cn (L.X.); 3Technology on Green-Eco-Healthy Animal Husbandry of Zhejiang Province, College of Animal Science and Technology, College of Veterinary Medicine, Zhejiang A&F University, Hangzhou 311300, China; rationalwang@163.com

**Keywords:** xylanase, growth performance, digestive absorption, intestinal barrier, metabolism

## Abstract

**Simple Summary:**

There has been increased cost pressure with the escalating prices of maize, and as such, wheat has gained attention as a potential alternative in animal husbandry. Wheat within non-starch polysaccharides (NSPs) with anti-nutritional components is worth our attention. Xylanase, as one of the non-starch polysaccharide enzymes, induces a positive influence on the growth performance, intestinal health, and microbiota of broilers. Feeding them a wheat-based diet with xylanase increased the activities of duodenal endogenous enzymes and enhanced the physical barrier of the intestines. Moreover, broilers fed xylanase through a wheat-based diet showed modulated cecal microflora; increased propionic acid produced by specific bacteria; and a changed serum metabolome, including histidine, cysteine, methionine, and other amino acid pathways. Based on the present study, xylanase supplementation is one potential high-wheat diet that could replace corn, playing an ideal role in poultry production.

**Abstract:**

This study aimed to investigate the effects of different levels of xylanase supplementation in a wheat-based diet on growth performance, short-chain fatty acids, intestinal health, microbial composition, and serum metabolism. A total of 1200 male chicks were randomly assigned to four wheat-based diet treatments: Group C (adding 0 mg/kg of xylanase), Group L (adding 50 mg/kg of xylanase), Group M (adding 100 mg/kg of xylanase), and Group H (adding 150 mg/kg of xylanase). The experiment lasted for 56 days. The results indicated that Group H broilers experienced a decreased feed-to-gain ratio throughout the study period. Additionally, dietary supplementation with xylanase led to an increase in the physical barrier, as indicated by increased VH and VH/CD in the gut (*p* < 0.05). Furthermore, levels of D-lactic acid and endotoxin were reduced. Xylanase supplementation also increased the abundance of Muc-2, ZO-1, and Occludin (*p* < 0.05). Moreover, xylanase supplementation enhanced the activity of sucrase and maltase in the duodenum (*p* < 0.05), which may be attributable to the upregulation of the abundance of SI and MGA (*p* < 0.05). Furthermore, xylanase addition promoted propionic acid produced by specific bacteria, such as *Phascolarctobacterium*, and influenced the microbial composition to some extent, promoting intestinal health. Additionally, 150 mg/kg of xylanase supplementation increased the amino acid, peptide, and carbohydrate content and upregulated the metabolism of amino acids related to histidine, cysteine, methionine, and other pathways (*p* < 0.05). These findings suggest adequate xylanase supplementation can enhance nutritional digestibility and absorption, improve growth performance, stimulate endogenous enzyme activity, optimize intestinal morphology and barrier function, and positively influence acid-producing bacteria and amino acid metabolic pathways.

## 1. Introduction

The feed gain ratio (F/G) is a crucial economic indicator in animal husbandry, as feed costs account for up to 70% of overall production expenses [1]. With the escalating prices of maize and soybean, there is increased cost pressure, resulting in reduced benefits in the current market scenario [2,3]. As a result, plant-based feeds such as wheat have gained attention as potential alternatives in animal husbandry. Wheat offers advantages such as high yield, affordability, and suitable protein and amino acid content [4,5,6].

It is important to note that wheat contains many anti-nutritional components, particularly non-starch polysaccharides (NSPs). One specific type of NSP found in wheat is arabinoxylan, which can undergo biological hydrolysis. This process increases intestinal viscosity and slows down the passage rate of chyme, resulting in reduced feed intake and digestibility. Arabinoxylan interferes with the formation of enzyme–substrate complexes and exerts strong negative effects on nutrient utilization [7,8]. Considering the detrimental effects of NSPs, using non-starch polysaccharide-degrading enzymes (NSPases) in feed formulations has become widespread and plays a crucial role [9]. NSPases enhance the digestibility of monogastric animals’ diets and release soluble fiber from the insoluble matrix. Additionally, they depolymerize viscous fibers into smaller carbohydrates, providing nutritional benefits [10].

Gut health plays a crucial role in the digestion and absorption of nutrients. Once the nutrients from the feed are broken down and absorbed by the digestive tract, they enter cells and are utilized to produce various compounds required by the body. Any undigested and unabsorbed nutrients undergo fermentation by the microbiota in the terminal part of the intestinal tract, resulting in the production of various metabolites that enter the circulation [11,12]. These microbiota-derived metabolites have regulatory functions in multiple aspects of host cell activity. For instance, bacteria translate fibers into short-chain fatty acids (SCFAs), the microbiome converts bile acids from the liver, and the microbiota transforms bioactive molecules derived from aromatic amino acids like tryptophan (Trp), all of which can subsequently impact the host [13].

Nevertheless, the application of xylanase in broilers is limited to intestinal microorganisms, and from the perspective of digestion and absorption, it is less understood regarding body metabolism. Therefore, our study aimed to assess the effects of adding xylanase to a wheat-based diet on growth performance, endogenous enzyme activity, and intestinal morphology in broilers. Additionally, we sought to explore the impact of xylanase supplementation on the intestinal barrier, SCFAs, microbiota composition, and the production of specific metabolites. Overall, we aimed to provide more comprehensive evidence regarding the metabolic improvements associated with xylanase supplementation.

## 2. Materials and Methods

### 2.1. Ethics Statement

This study adhered to the animal welfare guidelines set by China and received approval from the Animal Care and Welfare Committee, as well as the Scientific Ethical Committee, of Zhejiang University (No. ZJU20220310).

### 2.2. Design and Sample Collection

Xylanases were provided by DSM Co. Ltd. (Shanghai, China), and their activity was determined using the standard curve of a certified Ronozyme WX xylanase standard (2000 FXU/g). The experiment involved 1200 1-day-old male yellow-broiler chicks (initial BW = 33.29 ± 0.27 g). After placement at the farm (Xing Jian Culture-Farm, Jiaxing, China), the chicks were then divided into four groups, each comprising six replicates (50 chicks per replicate). They were housed on a floor covered with 5 cm deep wood shavings; each replicate occupied an area of 10 m^2^, received 24 h of continuous light, received barrel-feeding pellet feed, and was provided with random access to food and water. The temperature in the first week began at 35 °C in the brooding period and dropped to roughly 32 degrees at the age of 7 days, and then, the temperature was gradually reduced by 2–3 °C per week until reaching about 22 °C. The feeding strategy for the chicks included the following diets: the control group (Group C) received a wheat-based diet, Group L received a wheat-based diet with 50 mg/kg of xylanase in the feed, Group M received a wheat-based diet with 100 mg/kg of xylanase in the feed, and Group H received a wheat-based diet with 150 mg/kg of xylanase in the feed. The wheat-meal-based diet was formulated according to the nutritional requirements of broiler chickens as recommended in NY/T33-2004 (Table 1). At 21 and 56 days, all chicks were individually weighed within each replicate after an 8 h fasting period. The feed consumption and body weight of the chicks were recorded, and growth performance parameters were calculated for the starter period, grower period, and overall period.

### 2.3. Sample Collection and Storage

On day 56, two birds of intermediate weight were selected from each replicate, weighed, and slaughtered. Blood was taken from the underwing vein and allowed to stand at 37 °C for 2 h to facilitate separation. After centrifugation at 4000 rpm for 5 min, the serum was carefully collected and stored at −80 °C. A segment about 0.5 cm from the mid-duodenum and mid-jejunum was collected, gently rinsed with PBS (pH = 7.4), and then fixed in 4% paraformaldehyde solution. The cecal contents were scraped off meticulously using a clean blade and stored at −80 °C. Finally, the remaining gut segments were also preserved at −80 °C for future analysis.

### 2.4. Intestinal Index Determinations

The serum endotoxin levels (E039-1-1) were measured using a spectrophotometer from the Thermo Electron Corporation. Furthermore, the serum levels of D-lactic acid (H263) were determined using ELISA with the Infinite M200 Pro NanoQuant™ from Tecan. Additionally, the activities of lipase (A054-2-1), trypsin (A080-2-2), maltase (A082-3-1), sucrase (A082-2-1), and amylase (C016-1-1) in the duodenal mucosa were measured using the Infinite M200 Pro NanoQuant™. The above indicators were determined using assay kits following the instructions provided by the Nanjing Jiancheng Bioengineering Institute manufacturer (Nanjing, China).

### 2.5. Morphology Detection

Formaldehyde-fixed intestinal segments were dehydrated, embedded, and cut into 5 µm thick sections. Then, they were stained with hematoxylin–eosin for visualization. Goblet epithelial cells in the jejunum were identified using Alcian Blue and Periodic Acid Schiff (AB-PAS) staining. The villus height and crypt depth measurements were performed using the Image Pro Plus 6.0 software (Media Cybernetics, Rockville, MD, USA) of MediaCybernetics. Photomicrographs were captured using a Nikon ECLIPSE Ci Series microscope (Nikon Instruments Inc., Tokyo, Japan).

### 2.6. Gene Expression Analysis

The expression of disaccharidase genes, sucrase–isomaltase (SI), and maltase–glucoamylase (MGA) in the duodenal mucosa samples was determined. The expression of Muc-2, Occludin, Notch, and Claudin-1 in the duodenal and jejunal mucosa samples was also examined. A TRIzol reagent kit was used to extract total RNA from samples. RNA was reverse-transcribed into complementary DNA (cDNA) using the PrimeScript^TM^ RT Reagent Kit (Takara, Dalian, China). Quantitative real-time PCR (qRT-PCR) was performed using TB Green^®^ Premix Ex Taq ^TM^ (Takara, Dalian, China) and was conducted with a CFX96 Real-Time PCR Detection System from Bio-Rad using specific primers (Table 2). Each kit was operated following the instructions provided by TaKaRa Biotechnology. Furthermore, we referred to the 2^–ΔΔCt^ method of Livak and Schmittgen [14] for calculations.

### 2.7. Short-Chain Fatty Acids (SCFAs)

SCFAs were analyzed with gas chromatography, as described by Yang et al. [15]. Briefly, 1 g of cecal chyme was dissolved in deionized water and mixed thoroughly. Mixture was centrifuged at 12,000 rpm under 4 °C for 10 min to obtain the supernatant. Then, supernatant was vortexed and allowed to rest for 30 min to acquire another supernatant. Next, 1 mL of the new supernatant was mixed with 0.2 mL of 25% phosphoric acid. The mixed liquid stood at 4 °C for 30 min before being centrifuged. The final obtained supernatant was estimated using the appropriate machines.

### 2.8. Gut Microbiota Analysis

DNA extraction of microbiota from cecal chyme was performed, and the sequencing was conducted by Majorbio Bio-Pharm Technology. OTUs were clustered using 97% similarity, and sequences were identified and filtered accordingly. The alpha and beta diversity were calculated and performed by QIIME1. The sequence alignment was carried out using Blast, and sequences were annotated using the SILVA database. The figures were generated and visualized using R (version 3.3.1).

### 2.9. Metabolite Extraction and Analysis

The metabolomes in the serum samples were analyzed at Majorbio. We ensured that the features of the serum metabolites in the samples retained at least 80% of the data. Each metabolite underwent a normalization process, and the analysis was performed using log-transformed data to determine any significant differences. The metabolic features were estimated according to the Human Metabolome Database.

In line with the method described by Zhu et al. [16], the metabolic changes between groups were analyzed using Partial Least Squares Discriminant Analysis (PLS-DA) and Orthogonal Partial Least Squares Discriminant Analysis (OPLS-DA). Within the OPLS-DA model, the Variable Importance in the Projection (VIP) was calculated to identify the most influential metabolites contributing to the group differences. Furthermore, enrichment analysis of pathways associated with the identified metabolites was performed based on the Kyoto Encyclopedia of Genes and Genomes (KEGG).

### 2.10. Statistical Analysis

Data were analyzed using the SPSS statistical software (version 25) and presented as the mean value, and the standard error of the mean (SEM) was calculated. One-way ANOVA was performed to assess the differences between groups, and Tukey’s post hoc test was employed for post hoc comparisons. Differences between treatments were evaluated for significance at *p* < 0.05. GraphPad Prism (version 8) was utilized to generate the figures.

## 3. Results

### 3.1. Broiler Performance

Compared with Group C, the F/G of broilers decreased in Group M during the starter periods and declined in Group H during the overall periods (*p* < 0.05), and the dose group of 150 mg/kg showed a decreasing trend in F/G in the grower period (*p* = 0.082). Groups without differences in BW, ADG, and ADFI are shown in Table 3. These results show that supplementation with xylanase led to a significant decrease in F/G, and with a higher dose of xylanase, there was more improvement in feed conversion efficiency in the grower period.

### 3.2. Major Digestive Enzyme Activities

The activities of sucrose, maltase, and amylase were significantly increased in groups M and H (*p* < 0.05), as shown in Figure 1, compared with Group C. However, no significant differences were observed in the activities of lipase and trypsin among these groups (*p* > 0.05).

### 3.3. Intestinal Permeability

As can be seen in Figure 2, supplementation with xylanase at doses of 100 mg/kg and 150 mg/kg resulted in a significant decrease in the concentration of D-lactate in the duodenum, indicating a reduction in intestinal permeability (*p* < 0.05). Additionally, the endotoxin levels in the duodenum significantly decreased with all supplemented doses of xylanase (*p* < 0.05).

### 3.4. Intestinal Histomorphology

Groups M and H showed a significant decrease in CD and an increase in VH, as well as an increased VH/CD ratio in the duodenum (*p* < 0.05). Furthermore, supplementation with xylanase increased the VH and the VH/CD ratio compared with Group C in the jejunum (*p* < 0.05). Additionally, broilers in Groups M and H exhibited a higher abundance of goblet cells in the jejunum than in Group C, indicating that xylanase supplementation stimulated goblet cell counts in broilers (Figure 3A).

### 3.5. Intestinal Abundance of Relative mRNA

The mRNA levels of small intestine mucosal digestive absorption and mechanical barrier-related genes were evaluated (Figure 4). Xylanase supplementation resulted in the increased expression of glycol-enzymes (SI and MGA) and tight junction proteins (Occludin and ZO-1) in the duodenum, enhanced the mRNA levels of mucin and Occludin, and decreased the levels of Notch in the jejunum (*p* < 0.05). Furthermore, xylanase supplementation showed a tendency to increase mucin expression in the duodenum and ZO-1 expression in the jejunum.

### 3.6. Short-Chain Fatty Acid

The results showed that xylanase supplementation at 150 mg/kg (Group H) yielded optimal effects on broilers. Therefore, the subsequent assays focused on comparing Groups C and H. As shown in Figure 5, there were no significant differences in the content of acetic, butyric, and valeric acid; isobutyric acid; and total volatile fatty acids in the cecal contents between groups (*p* > 0.05). However, the propionic acid concentration was higher in Group H compared with Group C (*p* < 0.05).

### 3.7. Cecal Microbiota Analysis

The alpha diversity of the microbiota was assessed at the OTU level, and the results for six alpha diversity indexes are shown in Figure 6A. The results indicated no significant differences in alpha diversity. However, when examining the results of the principal component analysis and NMDS (Figure 6B), a clear separation between the groups could be observed, suggesting that Group H, which received xylanase supplementation at 150 mg/kg, had a distinct microbiota structure compared with the control group. Furthermore, LEfSe multilevel species (Figure 6C) analysis showed that *Barnesiella* and *Fournierella* could be characterized as the microbiota biomarkers of Group C, while *Verrucomicrobiota* could be recognized as a biomarker in Group H. Compared with Group C, supplementation with 150 mg/kg of xylanase improved the relative abundance of *Verrucomicrobiota*, *Phascolarctobacterium*, *Parabacteroides*, *Faecalibacterium*, *Lactobacillus*, and *Eubacterium_hallii_group* but decreased the level of *Barnesiella* (*p* < 0.05; Figure 6D).

### 3.8. Correlation Analysis

Spearman correlation analysis showed that isobutyric and *Parabactcroidcs* were positively correlated. Furthermore, propanoic acid positively correlated with *Phascolarctobacterium*, *Faccalibactcrium*, *Closrridia_vadinBB60*, and *UCG*–*014* (Figure 7).

### 3.9. Analysis of Metabolic and Pathways

The PLS-DA score plot (Figure 8A) revealed a clear separation between the groups, indicating distinct differences in metabolite profiles. This finding was further supported by the OPLS-DA score plot (Figure 8B). A total of 206 metabolites (62 downregulated and 144 upregulated) were identified in Group H compared with Group C based on criteria such as VIP > 1.0 and *p* < 0.05 (Figure 8C). The top 30 metabolites with the highest VIP values, including glutathione, carnosine, Threoninyl Proline, 4,5-dihydroorotic acid, and Epoxyganoderiol B, significantly increased in Group H (Figure 8E). However, the metabolite phosphocholine showed a decreasing trend. Pathway analysis revealed five significantly enriched pathways (*p* < 0.05): histidine; cysteine and methionine; glycine; serine and threonine; arginine biosynthesis; and beta-Alanine metabolism (Figure 8D(a)). Within these pathways, 15 important metabolites, including glutathione and carnosine, were identified (Figure 8D(b)). It is noteworthy that both carnosine and glutathione were among the top 30 VIP metabolites, indicating their potential importance in the metabolic changes observed in Group H.

## 4. Discussion

Earlier studies have demonstrated that supplementing NSP-rich diets with exogenous enzymes can benefit the host’s health [17]. For example, the supplementation of xylanase in wheat-based diets has been documented to improve feed conversion ratios and enhance growth performance. It also reduces chyme viscosity, increases the liberation of surplus nutrients for animals, and promotes intestinal health [18,19,20]. In line with these findings, our results indicate that xylanase supplementation significantly decreases the F/G ratio in a dose-dependent manner throughout the overall period.

Variations in the digestive tract can indicate changes in the capacity for digestion and absorption when different diets are provided [21]. Liu et al. [22] demonstrated that adding xylanase increases the activities of trypsin and amylase. Consistent with these findings, our study found that xylanase supplementation significantly increased the activities of endogenous enzymes such as amylase, sucrase, and maltase. Upon the arrival of starch in the duodenum, the activity of amylase on the mucosal surface of the intestine breaks down starch into dextrin, maltose, and other disaccharides, which are further converted into monosaccharides for direct absorption by the intestinal villi [23]. Regarding disaccharidases, we further examined gene expression and found that the upregulation of disaccharide genes may be associated with increased monosaccharide content, which activates sugar transporter receptors in intestinal epithelial cells, leading to the upregulation of upstream gene expressions related to external nutrient perception and the subsequent regulation of downstream disaccharidases. We hypothesize that this occurs due to the enhanced hydrolysis of non-starch polysaccharides by xylanase, increasing the substrate for endogenous enzymes. Consequently, the concentration of endogenous enzymes increases, providing optimal conditions for enzymatic reactions and improving the utilization of nutrients.

The intestinal epithelial cells play a considerable role in maintaining intestinal homeostasis by acting as physical and chemical barriers separating tissue from symbiotic bacteria. The mucous layer not only aids in the digestion of intestinal contents but also serves as a chemical defense line [24]. Kim and Ho [25] highlighted that reducing the Notch pathway can promote the differentiation of epithelial cells into secretory lineage cells. We observed that xylanase supplementation significantly decreased Notch gene expression, increased the number of goblet cells, and upregulated Muc-2 expression, which helps protect the mucosal barrier. Assessing intestinal development through intestinal morphology has become a reliable biomarker considering its role as a physical barrier within the gut, and it should be evaluated based on villi length (VH), crypt depth (CD), and permeability [26]. Previous research [27] has uncovered that supplementation with xylanase in different basic diets could improve the VH of the small intestine. Similarly, the addition of xylanase can enhance the VH and the VH/CD ratio, with increasing levels of xylanase supplemented in our study. The increase in VH could be attributed to a larger absorption surface and faster epithelial cell turnover [28], and a higher VH/CD may indicate a mature and active functional epithelium. Additionally, enzyme supplementation may have increased the relative abundance of tight junction proteins and reduced intestinal permeability in our research. These results provide evidence for enhanced nutrient absorption capacity and protective barrier function in the intestine.

Despite having a shorter intestinal transport and digestion time than mammals, poultry maintains a high digestive efficiency. Microbial colonization in the poultry gastrointestinal tract significantly influences the host’s health [29]. In our research, xylanase addition increased *Verrucomicrobiota* at the phylum level. Recent studies have documented that *Verrucomicrobiota* possesses enough glycoside hydrolase genes to degrade carbohydrates so that they release monosaccharides and amino acids that power other beneficial bacteria [30]. *Akkermansia muciniphila* is the most representative bacterium in *Verrucomicrobiota*, playing a role in mucin degradation regulation to protect the mucosal layer, promoting the production of prebiotics and SCFAs, and contributing to intestinal health and nutrient absorption [31]. Our study demonstrated that xylanase addition could promote the improvement of beneficial bacteria and play a role in intestinal protection. Consistently, it has been reported that adding xylanase can improve the microflora structure of broilers. By promoting the generation of propionate, acetic, and butyrate acid in beneficial bacteria, such as *Phascolarctobacterium*, *Parabacteroides*, *Faecalibacterium*, *Lactobacillus*, *Eubacterium_hallii_group*, and others, the content of butyrate-producing beneficial bacteria *Barnesiella* and *Fournierella* [32,33,34] was decreased. These findings indicate that the addition of xylanase could change the microflora structure to a certain extent and promote the production of SCFAs within a reasonable range. SCFAs play various roles, including promoting mucin production, regulating gene expression, stimulating enterocyte proliferation, and maintaining intestinal stability [35]. Previous studies have shown that adding xylanase to diets can increase propionic acid, isobutyric acid, butyric acid, and isovalerate [36]. Furthermore, Morgan et al. [37] indicated that dietary xylanase supplementation in laying hens could significantly promote acetic acid production. In addition, Singh et al. [38] documented enhanced acetate and total SCFA production after adding xylanase to corn–soybean meal basic diets. In our study, the contents of propionate, isobutyric, and total volatile fatty acids in the xylanase-fed group were higher than in the control. Although our findings differ from previous studies, this discrepancy may be due to differences in animal breeds, diet compositions, and particle sizes. Additionally, Isayama et al. [39] demonstrated that propionate modulates the tight junction barrier by increasing the expression of selective adhesion molecules in the endothelial cell and enhancing barrier functions of Caco-2 cells, suggesting that propionic acid might play a considerable role in the integrity of the mucosal barrier and the xylanase-enhanced intestinal barrier, so reduced permeability might closely be related to propionic acid in this research. Prebiotics produced through the hydrolysis of non-starch polysaccharides by xylanase can affect the cecal microbiome and increase fermentation metabolites, promoting villi growth and mucosal health [40].

Our analysis of the association between flora and volatile fatty acids demonstrated that *Parabacteroides* was positively correlated with isobutyric acid production, consistent with Qiao et al. [41], who showed that *Parabacteroides* could produce isobutyric acid by promoting microbial degradation, thus enhancing intestinal branch chain amino acid catabolism. In addition, propanoic acid was positively correlated with *Phascolarctobacterium* and *Faccalibactcrium* in our research. These bacteria have been established as beneficial bacteria [42], with *Phascolarctobacterium* being an important propionic acid producer [43]. Propionic acid is generated by propionic bacteria from various carbon sources, including xylose and arabose [44]. On the other hand, xylanase can degrade non-starch polysaccharides into prebiotic disaccharides, which propionic bacteria metabolize to produce propionic acid.

Metabolomics provides a unique and direct insight into the functional activities of an organism [45]. In our study, six major metabolites in the group receiving xylanase supplementation were upregulated: N-Acetyl-β-glucosaminylamine, N-Acetylserotonin glucuronide, Sucrose, Pyrogallol-2-O-glucuronide, Aspartylglycosamine, and Glucosamine 6-phosphate. This finding suggests that the addition of glycol enzymes is associated with the efficient utilization of glucose in animals. Furthermore, the upregulation of metabolites such as Glucosamine 6-phosphate indicates that glycolytic metabolic processes, including the pentose phosphate pathway and glycolysis, are activated following xylanase supplementation. Conversely, we observed a reduction in four major glycerophosphocholine metabolites, which may indicate coordinated adjustments aimed at maintaining energetic balance. Carbohydrate metabolism plays a crucial role in amino acid metabolism, as gluconeogenesis intermediates provide carbon to synthesize amino acid carbon frameworks [46].

Amino acids, peptides, and analogs significantly impact metabolism regulation in broilers [47]. In our present study, metabolomics analysis revealed the presence of overlapping metabolites (carnosine and glutathione) among the top 30 VIP values and the 15 important differential amino acid metabolic pathway metabolites. Carnosine, a dipeptide composed of L-histidine and β-alanine, is a potent antioxidant that cleans up the reactive oxygen species in the body [48]. Furthermore, carnosine is known to protect tissues from damage caused by “second-wave” chemicals. On the other hand, glutathione, one of the most crucial and potent antioxidants in animals, helps lower oxidative stress by neutralizing free radicals [49]. Additionally, glutathione plays a critical role in optimizing various factors that contribute to overall well-being, including inflammation management and immune function support [50]. In summary, xylanase supplemented in diets led to the identification of various amino acids, peptides, and analogs. In broilers, the observed composition of these compounds was found to be more conducive to healthy growth than a basic diet.

## 5. Conclusions

This study revealed that dietary supplementation with appropriate xylanase in a wheat diet could effectively dissolve anti-nutrient factors in feed; enhance the endogenous enzyme activity of the duodenum; promote intestinal acidophilic bacteria proliferation and the production of beneficial metabolites; improve intestinal development and intestinal barrier function; facilitate feed nutrient digestion and absorption; and achieve comprehensive effects by improving feed utilization. Additionally, our discovery suggests that xylanase supplementation is related to amino acid metabolic pathways, which could promote more beneficial metabolites for birds, ultimately achieving growth promotion performance.

## Figures and Tables

**Figure 1 animals-14-01182-f001:**
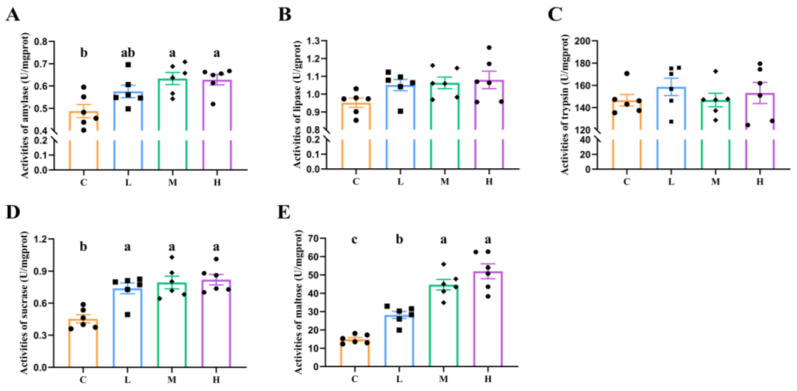
Effects of feeding xylanase to broilers on the major digestive enzyme activities in the duodenum. (**A**) Activities of amylase; (**B**) activities of lipase; (**C**) activities of trypsin; (**D**) activities of sucrase; (**E**) activities of maltose. Each group contained six biological replicates, data are shown as mean ± SEM, and a, b, c means with different superscripts differ significantly (*p* < 0.05). Abbreviations: C (Group C, control group), L (Group L, 50 mg/kg of xylanase in feed), M (Group M, 100 mg/kg of xylanase in feed), and H (Group H, 150 mg/kg of xylanase in feed).

**Figure 2 animals-14-01182-f002:**
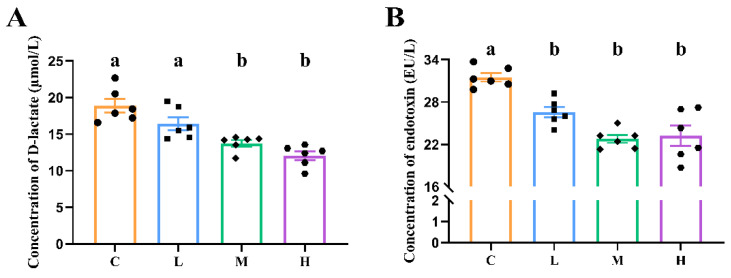
Effects of feeding xylanase to broilers on intestinal permeability. (**A**) Concentration of D-lactate; (**B**) concentration of endotoxin. Each group contained six biological replicates, data are shown as mean ± SEM, and means denoted by letter a and b indicate significant differences between treatments (*p* < 0.05). Abbreviations: C (Group C, control group), L (Group L, 50 mg/kg of xylanase in feed), M (Group M, 100 mg/kg of xylanase in feed), and H (Group H, 150 mg/kg of xylanase in feed).

**Figure 3 animals-14-01182-f003:**
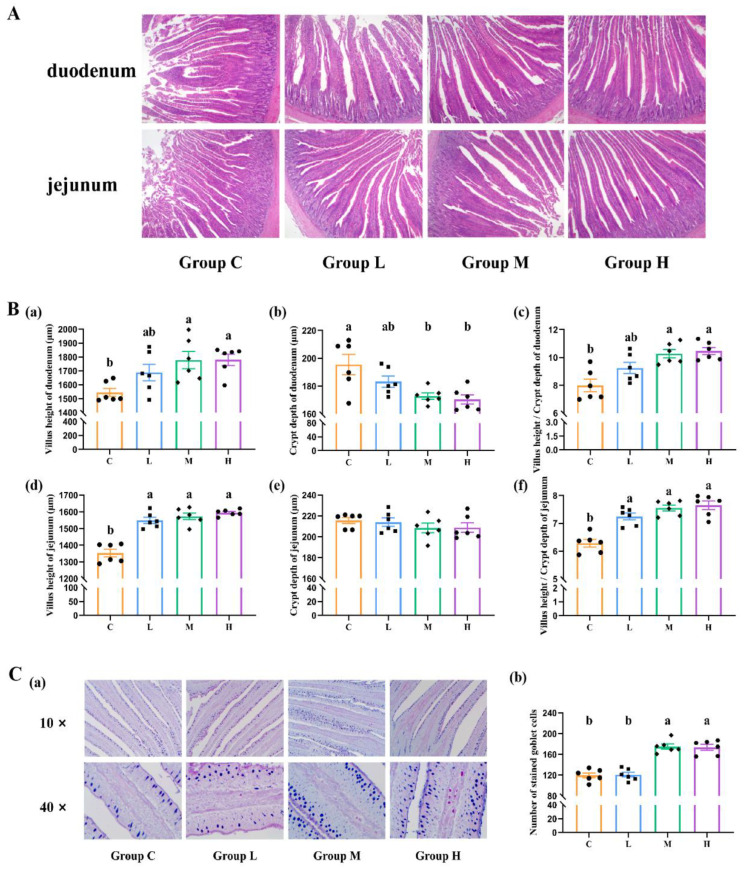
Effects of feeding xylanase to broilers on intestinal health. (**A**) Light microscopy of cross-sections of the duodenum and jejunum (40×). (**B**) Morphological structure of duodenum villus epithelium, villus height (**a**), crypt depth (**b**), and villus height-to-crypt depth ratio (**c**). Morphological structure of jejunum villus epithelium, villus height (**d**), crypt depth (**e**), and villus height-to-crypt depth ratio (**f**). (**C**) Light microscopy of cross-sections of the jejunum goblet cell (**a**) and the number of goblet cells in the jejunum mucosa (**b**). Each group contained six biological replicates, data are shown as mean ± SEM, and means denoted by letter a and b indicate significant differences between treatments (*p* < 0.05). Abbreviations: C (Group C, control group), L (Group L, 50 mg/kg of xylanase in feed), M (Group M, 100 mg/kg of xylanase in feed), and H (Group H, 150 mg/kg of xylanase in feed).

**Figure 4 animals-14-01182-f004:**
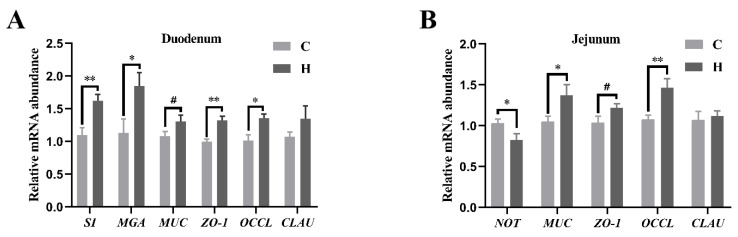
Effects of feeding xylanase to broilers on relative mRNA abundance in the gut. (**A**) Changes in relative mRNA abundance in the duodenum; (**B**) changes in relative mRNA abundance in the jejunum. The *t*-test was employed for post hoc comparisons. Each group contained six biological replicates, data were shown as mean ± SEM, and means denoted by different symbols indicate significant differences between treatments (*p* < 0.05); ^#^: 0.05 < *p* < 0.1, *: *p* < 0.05, and **: *p* < 0.01. Abbreviations: SI, sucrase–isomaltase; MGA, Maltase Glucoamylase; NOT, Notch; MUC, mucin; OCCL, Occludin; CLAU, Claudin; C (Group C, control group), and H (Group H, 150 mg/kg of xylanase in feed).

**Figure 5 animals-14-01182-f005:**
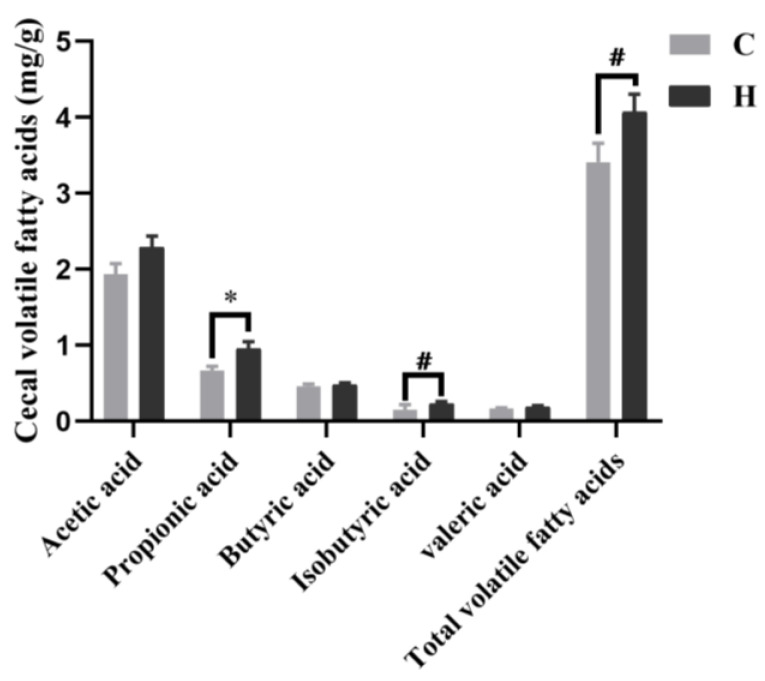
Effects of feeding xylanase to broilers on short-chain fatty acids. The *t*-test was employed for post hoc comparisons. Each group contained six biological replicates, and data are shown as mean ± SEM. Abbreviations: C (Group C, control group) and H (Group H, 150 mg/kg of xylanase in feed); ^#^: 0.05 < *p* < 0.1, and *: *p* < 0.05.

**Figure 6 animals-14-01182-f006:**
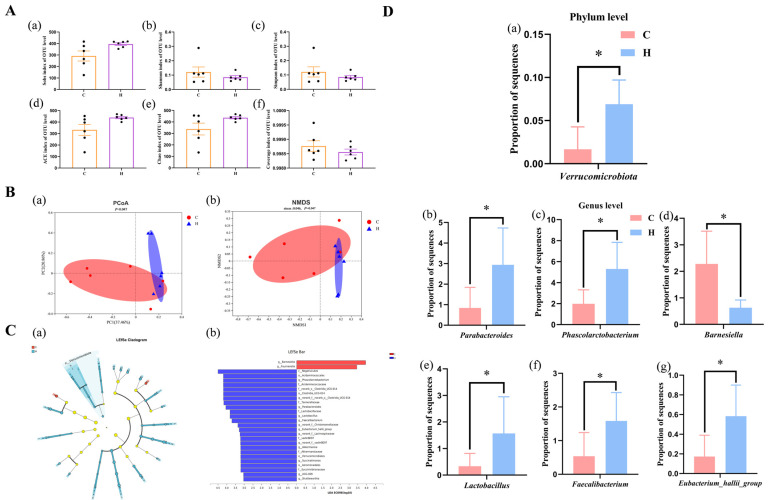
Effects of feeding xylanase to broilers on the composition of the gut microbiota. (**A**) α-diversity of the (**a**) Sobs, (**b**) Shannon, (**c**) Simpson, (**d**) ACE, (**e**) Chao and (**f**) Coverage index of OUT level; (**B**) β-diversity assessed by PcoA (**a**) and NMDS (**b**) analysis; (**C**) LEfSe analysis (LDA > 2.5) of Cladogram (**a**) and Bar (**b**); (**D**) column diagram of the relative abundance of *Verrucomicrobita* at the Phylum level (**a**) and the relative abundances of *Parabacteroides* (**b**), *Phascolarctobacterium* (**c**), *Barnesiella* (**d**), *Lactobacillus* (**e**), *Faeculibacterium* (**f**), and *Eubacterium_hallii_group* (**g**) at the genus level. Each plot represents one sample (*n* = 6). Abbreviations: C (Group C, control group) and H (Group H, 150 mg/kg of xylanase in feed). *: *p* < 0.05.

**Figure 7 animals-14-01182-f007:**
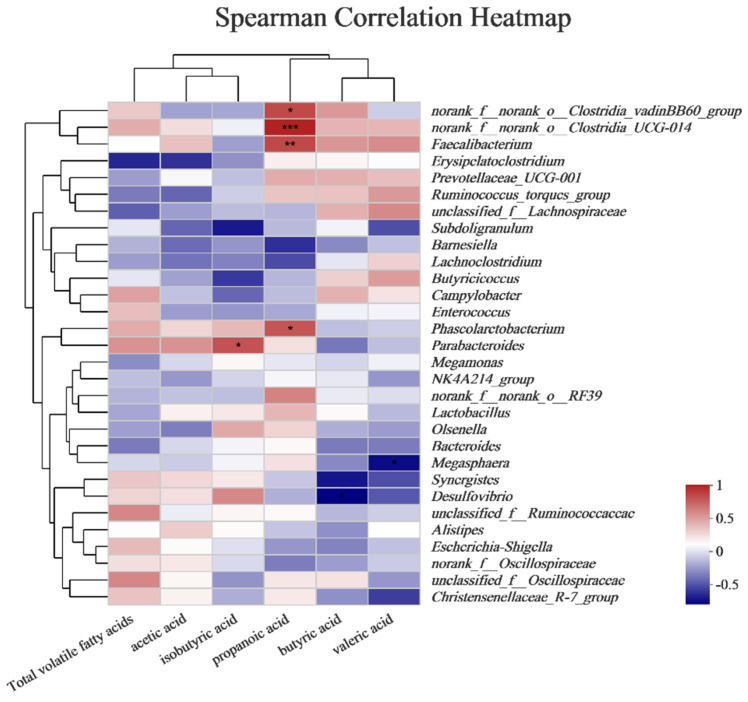
Correlation analysis of bacterial flora and volatile fatty acids. Each group contained six biological replicates. Abbreviations: *: 0.01 < *p* < 0.05, **: 0.001 < *p* < 0.01, and ***: *p* < 0.001.

**Figure 8 animals-14-01182-f008:**
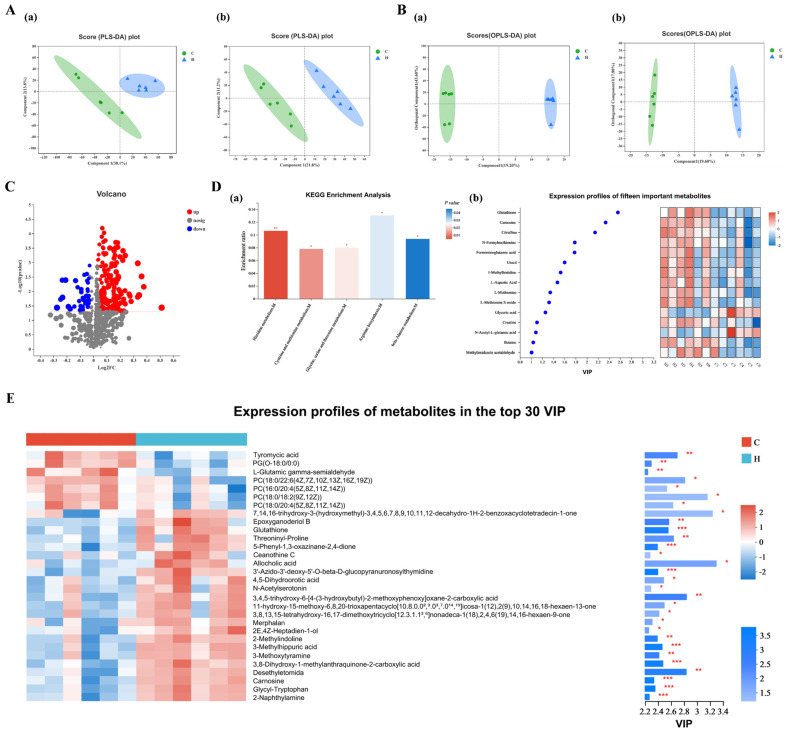
Xylanase changed the serum metabolic pathways and serum metabolites. (**A**) PLS-DA score plots among the samples of positive (**a**) and negative (**b**) ions; (**B**) OPLS-DA score plots among the samples of positive (**a**) and negative (**b**) ions; (**C**) volcano plot showing increased (red) and decreased (blue) levels of serum metabolic ions, with gray circles indicating no significant changes; (**D**) chart of KEGG enrichment analysis of the differential metabolism (**a**) and the corresponding fifteen important metabolites (**b**); (**E**) expression profile and bar charts of the VIP of the differential metabolites between two groups (in the top 30 VIP values). Each group contained six biological replicates. Abbreviations: C (Group C, control group) and H (Group H, 150 mg/kg of xylanase in feed). Abbreviations: *: 0.01 < *p* < 0.05, **: 0.001 < *p* < 0.01, and ***: *p* < 0.001.

**Table 1 animals-14-01182-t001:** Composition and nutrient levels of the basal diets for the broilers.

Items	1 to 21 Days	22 to 56 Days
Ingredients (%)		
Wheat	40.00	40.00
Corn	23.55	32.80
Soybean Meal (CP44%)	24.50	14.20
CGM (CP60%)	5.00	7.00
Soybean Oil	2.55	2.20
NaCl	0.30	0.30
CaHPO_4_	0.88	0.71
Limestone	1.74	1.50
Zeolite	0.48	0.29
Premix ^1^	1.00	1.00
Total	100.0	100.0
Nutrient levels ^2^ (%)		
ME (MJ/kg)	12.37	12.78
CP	21.00	18.52
Ca	0.90	0.76
TP	0.59	0.52
Lys	1.15	0.98
Met	0.49	0.42
Met+Cys	0.85	0.76
Thr	0.72	0.62
Digestible Lys	1.06	0.91
Digestible Met	0.47	0.40
Digestible Met+Cys	0.79	0.70
Digestible Thr	0.63	0.54
Try	0.22	0.18
Digestible Try	0.20	0.16

Abbreviations: ME, metabolizable energy; TP, total phosphorus; CP, crude protein; ME, metabolizable energy; CGM, corn gluten meal. ^1^ Supplied per kilogram of diet (mineral and vitamin premix): Se, 0.15 mg; I, 0.35 mg; Cu, 8 mg; Zn, 60 mg; Fe, 80 mg; Mn, 80 mg; vitamin B_1_, 3 mg; vitamin B_2_, 10.5 mg; vitamin B_6_, 4.2 mg; vitamin B_12_, 0.03 mg; folic acid, 1.5 mg; nicotinamide, 60 mg; D-calcium pantothenate, 18 mg; biotin, 0.225 mg; choline chloride, 1000 mg; vitamin K_3_, 3 mg; vitamin E, 36 mg; vitamin D_3_, 2700 IU; vitamin A, 9600 IU. ^2^ ME is the derived calculations, and other nutrients were determined through direct measurement.

**Table 2 animals-14-01182-t002:** Primer sequence of genes.

Genes	Forward	Reverse
β-actin	5′-GAGAAATTGTGCGTGACATCA-3′	5′-CCTGAACCTCTCATTGCCA-3′
SI	5′-GTGCTCCATGAGTTTGCACAAG-3′	5′-CAGCGGGCATTGGGTAAGTA-3′
MGA	5′-GGGACCAGATGACAAAATCCAT-3′	5′-CAGCGGGCATTGGGTAAGTA-3′
ZO-1	5′-CTTCAGGTGTTTCTCTTCCTCCTC-3′	5′-CTGTGGTTTCATGGCTGGATC-3′
MUC-2	5′-ATTGTGGTAACACCAACATTCATC-3′	5′-CTTTATAATGTCAGCACCAACTTCTC-3′
Occludin	5′-GAGCCCAGACTACCAAAGCAA-3′	5′-GCTTGATGTGGAAGAGCTTGTTG-3′
Claudin-1	5′-TGGCCACGTCATGGTATGG-3′	5′-AACGGGTGTGAAAGGGTCATAG-3′

Abbreviations: SI, sucrase–isomaltase; MGA, Maltase Glucoamylase.

**Table 3 animals-14-01182-t003:** Effects of xylanase on growth performance in broilers ^1^.

Items	Treatments ^2^				SEM	*p*-Value
	C	L	M	H		
**Starter (1~21 days)**						
1 day BW (g)	33.10	33.40	33.28	33.38	0.052	0.135
21 days BW (g)	424.8	433.0	448.2	426.8	3.455	0.078
ADFI (g/d)	30.98	30.98	31.59	30.68	0.153	0.233
ADG (g/d)	18.65	19.03	19.76	18.74	0.164	0.077
F/G	1.66 ^a^	1.63 ^ab^	1.60 ^b^	1.64 ^ab^	0.008	0.049
**Grower (22~56 days)**						
56 days BW (g)	1788	1788	1820	1802	12.38	0.818
ADFI (g/d)	94.36	93.19	94.02	92.97	0.746	0.911
ADG (g/d)	38.85	38.62	39.13	39.34	0.324	0.884
F/G	2.43	2.41	2.40	2.36	0.008	0.082
**Overall (1~56 days)**						
ADFI (g/d)	70.68	69.90	70.60	69.51	0.462	0.798
ADG (g/d)	31.34	31.34	31.91	31.58	0.221	0.818
F/G	2.26 ^a^	2.23 ^ab^	2.21 ^ab^	2.20 ^b^	0.008	0.042

^a,b^ Means with different superscripts differ significantly (*p* < 0.05) within the same row. ^1^ Each group is *n* = 6. ^2^ Abbreviations: Group C (control group), Group L (50 mg/kg of xylanase in feed), Group M (100 mg/kg of xylanase in feed), and Group H (150 mg/kg of xylanase in feed).

## Data Availability

The data that support the findings of this study are available upon request from Xiuan Zhan. The data and materials in this study are available from the corresponding authors upon reasonable request.

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
