# Peer review of "Xylanase Supplement Enhances the Growth Performance of Broiler by Modulating Serum Metabolism, Intestinal Health, Short-Chain Fatty Acid Composition, and Microbiota"

_animals, 2024, doi:10.3390/ani14081182_

Round 1

Reviewer 1 Report

Comments and Suggestions for Authors

Dear Authors,

Line 19: Group H was the only group that had significant effects on feed efficiency. Consequently, it would be unsuitable to include Group M in this context.

Lines 64-67: Numerous research are available that are relevant to your field of study. You should enhance the customization of your research by including unique aspects that distinguish it from previous research.

Lines 85 to 87: By what criteria did you establish the application dosages? Please provide more specific details.

Lines 78-91 include details on stocking density, lighting conditions, feeder characteristics, and drinker specifications.

Lines 78 to 91: Why did you ignore to assess the viability data? You have to explain it clearly, even in the absence of mortality.

Table 1: In general (1-56 days) you need to give body weight and body weight gain. Total feed consumption is more important than daily feed consumption.

Lines 248-249: The application had an effect only on propionic acid, its effect on others was insignificant. The sentence needs to be rewritten.

Figure 5:  Replace “propanoic acid” with “propionic acid”

Best regards,

Author Response

Responds to reviewer#1’s comments:

Line 19: Group H was the only group that had significant effects on feed efficiency. Consequently, it would be unsuitable to include Group M in this context.

Response: We are very sorry for our unclear describe. We have made major revisions for it. Results indicated that Group H broilers experienced a decreased feed-to-gain ratio throughout the study period.

Lines 64-67: Numerous research are available that are relevant to your field of study. You should enhance the customization of your research by including unique aspects that distinguish it from previous research.

Response: Thank you for your suggestion. We have made major revisions and the change in our manuscript following your suggestion marked yellow highlighting. 

Lines 85 to 87: By what criteria did you establish the application dosages? Please provide more specific details.

Response: Thank you for your comment. The application doses of the enzymes used in this trial were determined with reference to the basis of previous studies (Melo-Duran et al., 2021) and the enzyme activity of the company's novel xylanase.

Lines 78-91: include details on stocking density, lighting conditions, feeder characteristics, and drinker specifications.

Response: We are very sorry for our unclear describe. We have made major revisions for it. They were housed in floor covered with 5cm deep wood shavings,each replicate occupied an area of 60 m2, 24h continuous light, barrel feeding and provided randomly access to food and water.

Lines 78 to 91: Why did you ignore to assess the viability data? You have to explain it clearly, even in the absence of mortality.

Response: Thank you for your suggestion. We did not ignore to assess the viability data, the mortality of broilers in this study ranged from 2% to 3%, and was not included in the article based on previous studies showing that feeding the enzyme did not have a significant effect on broiler mortality.

Table 3: In general (1-56 days) you need to give body weight and body weight gain. Total feed consumption is more important than daily feed consumption.

Response: Thank you for your suggestion. More detailed information as follow.

Items

Treatments2

SEM

p-value

C

L

M

H

Overall (1~56d)

56d Body weight (g)

1788

1788

1802

1820

12.38

0.818

Body weight gain(g)

1,771.7

1,771.9

1,805.6

1,796.0

12.41

0.744

Total feed consumption(g)

3996.5

3951.6

3994.6

3952.4

23.83

0.865

ADFI(g/d)

70.68

69.90

70.60

69.51

0.462

0.798

ADG(g/d)

31.34

31.34

31.91

31.58

0.221

0.818

F/G

2.26a

2.23ab

2.21ab

2.20b

0.008

0.042

Lines 248-249: The application had an effect only on propionic acid, its effect on others was insignificant. The sentence needs to be rewritten. Figure 5: Replace “propanoic acid” with “propionic acid

Response: We are very sorry for our unclear describe. We have made major revisions for it. 

Reviewer 2 Report

Comments and Suggestions for Authors

Dear Authors, your interesting manuscript is very useful and made an important relationship between feed additives and gut health, however, for publishing I am asking for some corrections, which you can see in the attached file.

Best regards,

Author Response

Responds to reviewer#2’s comments:

Page 1 Line 34: please, choose other keywords, which do not appear in the title.

Response: Thank you for your suggestion. We have made the revisions for it. 

Page 2 Line 83: please, include a full description of the local which the broiler trial was conducted (university facilities? Name of it?), a full description of the pen size and housing density, as well as the light program and house temperatures used during the trial. Line 87: please, include the diet physical form, pellet, mash, crumbled.

Response: Thank you for your suggestion. We have made the revisions for it in lines 81 to 87. 

Page 3 Table 1: Did you use a coccidiostat product? If yes, please include.

Response: Thank you for your suggestion. A routine immunization procedure of broilers was conducted in this experiment. 

Page 3 Table 1 Line 100: please, be clearer by explaining. Are these values analyzed values? Did you analyze the enzyme in the diets?

Response: Thank you for your suggestion. These values was formulated according to the nutritional requirements of broiler chickens as recommended in NY/T33-2004. The values in the table are measured values. In addition, we do not analyze enzymes in the diet.

Line 102: please, delete the word from . Page 5

Response: Thank you for your suggestion. We have made the revisions for it.

Lines 168 to 172: please, provide if you test the data for normality prior to ANOVA and the full statistical model.

Response: Thank you for your suggestion. The data in this study were analyzed for ANOVA and complete statistics only when the conditions of independence, normality, and chi-square were met, we just didn't present these in the manuscript.

Page 8 Line 218: please, check the font size of the word xylanase. Pages 14 and 15

Response: Thank you for your suggestion. We have made the revisions for it.

Page 8 Lines 425 to 433: please, follow the Journal guidelines and write it in a separate section. 5. Conclusions .

Response: Thank you for your suggestion. We have made major revisions for it.

Reviewer 3 Report

Comments and Suggestions for Authors

Xylanase supplement enhanced the growth performance of broiler by modulating serum metabolism, intestinal health, short-chain fatty acid, and microbiota

Dear Authors,

The manuscript is interesting, and describes positive effect of xylanase supplementation on feed conversion ratio, intestinal barrier function and microbiome. There are some aspects to correct.

Below I add some suggestions helpful in this process:

Line 1

Title of manuscript is: Xylanase supplement enhanced the growth performance of broiler by modulating serum metabolism, intestinal health, short-chain fatty acid, and microbiota.

Maybe it is worth to consider if add: Xylanase supplement enhanced the growth performance of broiler by modulating serum metabolism, intestinal health, short-chain fatty acid composition, and microbiota (intestinal  microbiome).

Line 16

In text is sentence: ‘…A total of 1200 male chicks were ad-libitum assigned to…’, better is to change to ‘…A total of 1200 male chicks were randomly assigned to…’ (ad libitum is form of feed offered chickens when they have access to feed in every moment).

Line 17-308

In text is: 0mg/kg xylanase, space needed: 0 mg/kg xylanase

And so on with other values…

Line 22-294

In text of manuscript is P-value, must be p-value

And so on with in following lines…

Line 93

Table 1

In case of Soybean meal (% CP) could be useful information

In case of amino acids:

Is The (threonine), must be Thr

Try (tryptophane), must be Trp and Digestible Trp on the end of table.

Line 140-389

In text is: Yang et al [15], must be with dot: Yang et al. [15].

And so on line 160, …

Line 168

In text of manuscript is: ‘…and were reported as the standard error of the mean (SEM)…’, mean must be added ‘…and were presented as mean value, additionally  the standard error of the mean (SEM) was calculated’.

Line 170

Tukey’s post hoc test is enough (without t-test, because that suggest least significant difference test, important in the past, but is less precise in comparison with Tukey’s test).

In text is sentence: ‘…P < 0.05 was Statement of significance…”, maybe better will sounds: ‘…Differences between treatments were evaluated for significance at p < 0.05…’.

Line 236

Why without L and M treatments? In case of performance M treatment is one of the best results?

Maybe not statistically significant. But it is only suggestion.

Line 425

Lack of section: 5. Conclusions in manuscript.

Last paragraph can be added to this section or modified if required.

Line 451

Ordinal numbers without brackets must be used in Reference section

1.

2.

3.

50.

Author Response

Responds to reviewer#3’s comments:

Line 1: Title of manuscript is: Xylanase supplement enhanced the growth performance of broiler by modulating serum metabolism, intestinal health, short-chain fatty acid, and microbiota. Maybe it is worth to consider if add: Xylanase supplement enhanced the growth performance of broiler by modulating serum metabolism, intestinal health, short-chain fatty acid composition, and microbiota (intestinal microbiome).

Response: Thank you for your suggestion. We have made major revisions for it.

Line 16: In text is sentence: ‘…A total of 1200 male chicks were ad-libitum assigned to…’, better is to change to ‘…A total of 1200 male chicks were randomly assigned to…’ (ad libitum is form of feed offered chickens when they have access to feed in every moment).

Response: Thank you for your suggestion. We have made major revisions for it.

Line 17-308: In text is: 0mg/kg xylanase, space needed: 0 mg/kg xylanase And so on with other values…

Response: Thank you for your suggestion. We have made major revisions for it.

Line 22-294: In text of manuscript is P-value, must be p-value.And so on with in following lines…

Response: Thank you for your suggestion. We have made major revisions for it.

Line 93: Table 1 In case of Soybean meal (% CP) could be useful information. In case of amino acids: Is The (threonine), must be Thr; Try (tryptophane), must be Trp and Digestible Trp on the end of table.

Response: Thank you for your suggestion. We have made major revisions for it.

Line 140-389: In text is: Yang et al [15], must be with dot: Yang et al. [15]. And so on line 160, …

Response: Thank you for your suggestion. We have made the revisions for it.

Line 168: In text of manuscript is: ‘…and were reported as the standard error of the mean (SEM)…’, mean must be added ‘…and were presented as mean value, additionally  the standard error of the mean (SEM) was calculated’. Line 170: Tukey’s post hoc test is enough (without t-test, because that suggest least significant difference test, important in the past, but is less precise in comparison with Tukey’s test).In text is sentence: ‘…P < 0.05 was Statement of significance…”, maybe better will sounds: ‘…Differences between treatments were evaluated for significance at p < 0.05…’.

Response: Thank you for your suggestion. We have made major revisions for it.

Line 236: Why without L and M treatments? In case of performance M treatment is one of the best results? Maybe not statistically significant. But it is only suggestion.

Response: Thank you for your suggestion. We have made major revisions for it. We want to explore the mechanism of action from the best results. The dose of M (100 mg/kg) should have the best effect in the early growth stage, while the effect of group H is better in the late growth stage. From the perspective of the whole period, we chose group H for comparison.

Line 425:Lack of section: 5. Conclusions in manuscript. Last paragraph can be added to this section or modified if required.

Response: Thank you for your suggestion. We have made major revisions for it.

Line 451: Ordinal numbers without brackets must be used in Reference section

1.2.3.…50.

Response: Thank you for your suggestion. We have made major revisions for it.